

# GC Insights: Geoscience students' experience of writing academic poetry as an aid to their science education

Alice Wardle[1], Sam Illingworth[2]

[1]School of Psychology and Therapeutic Studies, Leeds Trinity University, Leeds, UK

[2]The Department of Learning & Teaching Enhancement, Edinburgh Napier University, Edinburgh, UK

*Correspondence to*: Alice Wardle (1906082@leedstrinity.ac.uk)

**Abstract**

The study presented here employs thematic analysis to explore geoscience students' experience of writing poetry as an aid to their science education. It was found that themes could be categorised as being related to either the 'Task Process' or 'Task Meaning'. The results of this study present evidence that writing poetry can aid geoscience students by making newly learned information more digestible, and therefore easier to memorise efficiently.

### 1. Introduction

Science and the humanities are often seen to be disparate fields of study. However, recently, there has been a desire to reignite the bond between science and the arts as a way to advance education, pedagogy, and communication between academic disciplines (Osbourn, 2006; Brown, 2019; Gurnon et al., 2013; Watts, 2001). It has also been argued that science journal articles are sometimes too esoteric, making it difficult for scientists from different areas of study to understand and communicate with each other. Communicating using poetic, creative language could

potentially ameliorate this difficulty (Osbourn, 2006, Illingworth, 2022).

In a study by Pollack and Korol (2013), undergraduate neuroscience students were challenged to write a haiku based on an excerpt from a textbook, a task that was found to require contemplation and lateral thinking. Inspired by this study, our research presents an introduction to exploring geoscience students' experience of writing academic poetry as an aid to their science education.

### 2. Method

A research advertisement was posted to social media platforms and participants were recruited using opportunity sampling. The inclusion criteria stated participants had to be a university student aged 18 or over, enrolled in a bachelor of science or master's degree in a geoscience-related subject, and have a university email address. In total 11 participants took part in this qualitative study.



Ethical approval for this study was given by the Leeds Trinity School of Psychology and Therapeutic Ethics
       Committee. Aside from preserving anonymity, one of the biggest ethical issues to consider was that writing poetry
       might trigger emotional effects in the participants, and as such we provided signposting to charities (Safe In Our
       World and Mind) who could provide support and mental health advice if required.

       A link was shared on social media platforms that directed participants to the survey, which was hosted by Jisc
Online Surveys. After reading the information sheet and accepting the consent form (see Appendix A), participants
       were shown a list of questions to complete concerning the university they attend, the name of the course they were
       enrolled in, and their university email address. Participants were asked to choose one out of a possible three text
       passages (labelled anonymously as "1", "2", and "3": The Royal Society, 2020a, para 2.; see Appendix B; The
       Royal Society, 2020b, para 3.; see Appendix C; The Royal Society, 2020c, para 2.; see Appendix D). Following this
selection, they were then invited to read their chosen passage of text and then write a haiku (which we defined here
       as a three-line poem of 17 syllables) based on the information they had just read. They were then asked four open-
       ended questions about their experience of writing the haiku:

       1. How did you find the experience of writing the poem?

       2. Did writing the poem affect your engagement with the science in any way?

3. Was a haiku an appropriate form to use?

       4. Have you experience of writing poetry before this exercise?

### 3.    Results & Discussion

       Thematic analysis was used to analyse the data in this study. According to Braun and Clarke (2012), a thematic
       analysis is a qualitative data analysis that extracts themes across a data set to discover meaningful patterns.
Important information in the text that relates to the research question is identified and summarised using a word or
       short phrase, which are called codes. There may be several codes that are very similar to one another and can
       collapse to form one subtheme or theme.

       The themes and subthemes extracted from the data within this study were associated with either the 'Task Process',
       which we define as focussing on the steps within the process of conducting the task, or the 'Task Meaning', which
focuses on the emotions and sense of meaning participants experienced while writing their haiku.

### 3.1  Task Process

       The themes that emerged from Task Process were found to be: 'Identification of significant information',
       'Distillation of information', and 'Metamorphosis of text' (see Fig. 1a). After reading the text extract, participants
       began identifying significant information from the text extract:

60           "A haiku is very short so I had to really get the core words out of the text"





Participant 1

Following this, they distilled the information, filtering out the information they deemed as least relevant:

"It made me think much more about how to distil the essential information"

Participant 1

And then they metamorphized the text from an academic to a poetic style:

"It is a good use to metamorphize a scientific situation form to monotone (dry) text to a vivid (appealing)"

Participant 8

Writing the haiku was also found to be an iterative process as opposed to a linear one:

"Even after I finished writing the poem I went back to change it after thinking there was a better piece of
information to put in the second line than what I originally put"

Participant 11

### 3.2 Task Meaning

The themes to emerge from 'Task Meaning' were found to be 'Enjoyable', 'Challenging', and 'Valuable'.
Furthermore, 'Challenging' is constructed from two subthemes: 'Frustrating' and 'Restricted' (see Fig. 1b).

The majority of the participants found the task to be enjoyable:

"It was a great experience! I always get warm fuzzies in my stomach when I'm able to join my creative and
scientific brains together"

Participant 4

The meaning and emotional experience of the task is interconnected with the Task Process. In this case, the
participant enjoyed transfiguring the text from its original academic style to something more vivacious.

As well as being an enjoyable task, most of the participants found the task to be challenging due to the restrictions
given.

Many participants found the use of the haiku an appropriate poetry form for this study and that the restrictions were
relevant:

"It was challenging to really boil down the concept, it was a great exercise in simplification and
summarization".





Participant 4

However, some participants found that the directions that had been given, to write a 17-syllable haiku with three lines, were too limiting. This was noticeable as some of the participants went beyond the syllable count. They may
have been unaware of their counting error, or the error could have been due to them feeling too constricted in what they could write:

"I found it limiting in terms of the amount of detail I could give".

Participant 10

When participants felt too restricted, it also caused them to feel frustrated:

"[I found it] frustrating at times"

Participant 8

Finally, it is important for students to find value in doing the task. Participants identified that it helped them to memorise the information within the larger text extract; instead of having to remember a large paragraph of information, they only needed to recall three short lines of a poem, which contained the most significant and
relevant pieces of information:

"I learned science I didn't know from the passage and now won't forget it because I made a fun mnemonic"

Participant 4



a)

b)

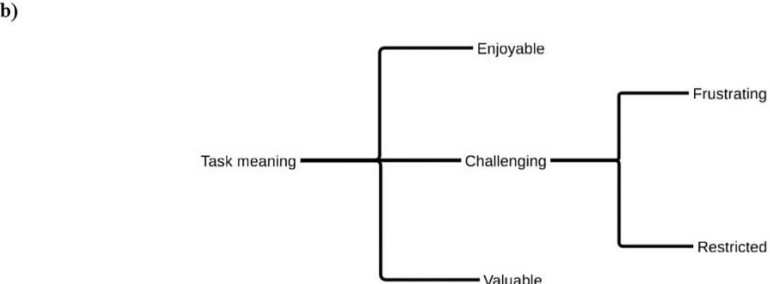

**Figure 1.** (**a**) Task Process thematic map. (**b**) Task Meaning thematic map

The results of this study support and expand on the findings of Pollack and Korol (2013), in which neuroscience students were asked to write a haiku based on a passage of text. Whereas their study focused on whether haiku can be used to describe and explain neuroscience concepts, this study focused on both the experience of constructing a haiku and the subjective experience of the students of doing it in relation to the geosciences.

## 4. Conclusion

This study explored whether writing poetry can be used by geoscience students as an aid to their science education. The thematic analysis conducted on the data confirmed that the experience of writing a haiku based on a text extract can be enjoyable, challenging, and valuable. However, tasks involving larger text extracts and alternate forms of poetry may have very different findings and result in a different subjective experience.

Though the questions were open-ended, it was possible for participants to give a simple binary response of 'Yes' or
'No' for three out of the four questions, which may have prevented more in-depth responses from participants. Future research may want to examine the experience of students studying different geoscience subjects and pose



different questions to explore different aspects of their experience, including if the task has made them consider writing poetry to aid their science education in the future. There was also a limited amount of data due to the small number of participants within this study; we hope to expand this to larger numbers in the future.

**Data availability**

The data of this study, which includes each participants' haiku and responses to the four questions, is available at https://osf.io/a24sz/?view_only=9c0914834c264e8490a5931422992dd6

**Author contributions**

AW and SI both designed and delivered the survey, with AW conducting the first analysis of the data. AW and SI
co-wrote the paper.

**Competing interests**

Sam Illingworth is the chief executive editor of *Geoscience Communication*.

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



**Appendices**

**Appendix A**

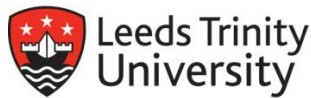

**Geoscience students' experience of writing academic poetry as an aid to their science education – Consent form**

If you are 18 years old or over and would like to consent to the use of your data as detailed in the information sheet then click the 'next' button below. You will be asked to fill in details about your university, the name of your

course, and your university email address. Following this, you will be shown a text passage about a topic within geoscience and asked to write a poem based on the information within the text. If you do not consent to the information above and would not like to participate in this study then please close the tab to this page.

Understand that you can leave the study at any point by exiting the tab to Jisc Online Surveys. If you do this before

completing the four questions at the end of the study then none of your data will be saved. If you would like to withdraw from the study upon completion of the four questions then please email Alice Wardle at 1906082@leedstrinity.ac.uk that you would like to do so. You will also need to send your unique response number, which will be shown to you at the end of the study.



## Appendix B

### Ocean Acidification text passage

CO2 dissolves in water to form a weak acid, and the oceans have absorbed about a third of the CO2 resulting from human activities, leading to a steady decrease in ocean pH levels. With increasing atmospheric CO2, this chemical balance will change even more during the next century. Laboratory and other experiments show that under high

CO2 and in more acidic waters, some marine species have misshapen shells and lower growth rates, although the effect varies among species. Acidification also alters the cycling of nutrients and many other elements and compounds in the ocean, and it is likely to shift the competitive advantage among species, with as-yet-to-be-determined impacts on marine ecosystems and the food web.



**Appendix C**

180                      **Climate change and weather events text passage**

A warming climate can contribute to the intensity of heat waves by increasing the chances of very hot days and nights. Climate warming also increases evaporation on land, which can worsen drought and create conditions more prone to wildfire and a longer wildfire season. A warming atmosphere is also associated with heavier precipitation events (rain and snowstorms) through increases in the air's capacity to hold moisture. El Niño events favour drought

in many tropical and subtropical land areas, while La Niña events promote wetter conditions in many places. These short-term and regional variations are expected to become more extreme in a warming climate.



**Appendix D**

**Sea ice extent in Arctic and Antarctic sea**

Some differences in seasonal sea ice extent between the Arctic and Antarctic are due to basic geography and its
influence on atmospheric and oceanic circulation. The Arctic is an ocean basin surrounded largely by mountainous
continental land masses, and Antarctica is a continent surrounded by ocean. In the Arctic, sea ice extent is limited by
the surrounding land masses. In the Southern Ocean winter, sea ice can expand freely into the surrounding ocean,
with its southern boundary set by the coastline of Antarctica. Because Antarctic sea ice forms at latitudes further
from the South Pole (and closer to the equator), less ice survives the summer. Sea ice extent in both poles changes
seasonally; however, longer-term variability in summer and winter ice extent is different in each hemisphere, due in
part to these basic geographical differences.