# Peer review of "GC Insights: Geoscience students' experience of writing academic poetry as an aid to their science education"

_EGUsphere, 2022_

## Author Response (AR1)

**Reply for Stephany Buenrostro Mazon**

**Thank you for taking the time to read this manuscript, and for providing helpful and specific feedback for how to improve this work. Below we have responded to all your comments and indicated how we have changed the manuscript to account for these changes (which for ease of use we have written in italics). Any line references refer to those provided in the Geoscience Communication Discussions preprint.**

Clarification in the writing of line 11: "It was found the themes could be categorized…". The authors could consider introducing the themes found in the study.

*Changes have now been made to the manuscript which introduce the themes found within the study.*

Line 12: "The results of this study present evidence that writing poetry can aid geoscience students…". The sentence would seem to suggest that the act of writing poetry itself is what aids students, as opposed to writing poetry from scientific texts. Consider re-wording.

*The sentence has been changed from "The results of this study present evidence that writing poetry can aid geoscience students…" to "The results of this study present evidence that writing haiku based on scientific text can aid geoscience students…". This change makes it more obvious that writing poetry based on scientific text is what aids their learning rather than writing generic poetry.*

Line 13: The last sentence states it was 'easier to memorise efficiently', however this was not part of the assessment. The survey did not assess information retention at a later date, rather it is based on a participant's comment. This conclusion without the context of the participant and survey therefore seems slightly misleading in the abstract.

*To avoid readers interpreting this sentence in the way you have described, "…and therefore easier to memorise efficiently" has been removed from the final sentence.*

The abstract is very short. I suggest to include the sample size, the survey as the data collection method, and results from the Task Meaning themes.

*We have added information about the sample size, survey as the data collection method, and the themes from both the Task Process and Task Meaning.*

The methods section is clear in how the instructions given to participating students. However, given the sample size was small (11 participants), the paper could be strengthen by including samples of the social media marketing material used and the

communication strategy (how many days was the call advertised? which channels? was it at a university level or through private social media accounts?). Social media analytics have shown that different channels are preferred by different age groups, and the predominant format of information (text, image, video) can also vary depending on the channel. This could all have an effect on which students saw the call, and could serve the authors and readers wanting to replicate the study to modify how they target their participants.

*The first sentence of the method has now been extended to include information about what social media platforms were used: "A research advertisement was posted to social media platforms, including the personal researchers' accounts on Twitter, Facebook, and LinkedIn, and shared by leaders of geography-related modules to students in the UK; participants were recruited using opportunity sampling." An appendix has also been added that presents the research advertisement used for this study.*

The authors could include a brief discussion on the possible bias of the participants in already having experience with poetry, and even using it in their personal life. From the 11 participants, 10 had previous experience with poetry, and 9 used it outside of their academic material.

*Due to the 1500 word count for a GC Insights study, we could not include all the information we would have liked, including information about the possible bias of the participants who have already have experience writing poetry. However, the bias has now been included as a limitation in the Conclusion: "There was also a limited amount of data due to the small number of participants within this study, with Participant 10 being the only person who did not have prior experience writing poetry."*

Could the call or the marketing strategy have affected this? For example, the #scicomm hashtag would be visible to twitter users active in and/or following science communication activities. This is relevant to the study as the aim is to explore the use of poetry in learning, and so it is important to consider the spectrum of students with different dispositions to transdisciplinarity and/or artistic tendencies in how they use and benefit from using poetry.

*Thank you for this suggestion, we agree that for future study's we could widen the scope further. However, the #scicomm was actually used (along with (#sciart) by the authors in their original advertisement.*

While the Results found haiku to be an adequate form of poetry to distill essential information, there was no clear motivation as to why the authors chose the Haiku for the study as opposed to other forms of poetry. The importance of the succinctness and the 'iterative process' of the haiku could be emphasized in methods.

*Due to the limited word count, unfortunately, a detailed explanation on why a haiku was chosen for the study is not possible. In the 'Introduction' section, however, as*

*was mentioned this study is inspired by the method of the Pollack and Korol (2013). In the 'Conclusion' section, it is now also proposed that other forms of poems are used in future studies. Due to the limited word count of a GC Insights article, a haiku is appropriate as the responses to the questions are more likely to also be succinct.*

The excerpt text assigned to participants are part of reports as information and educational material by the Royal Academy, rather than a purely scientific text, like a research article. The aim of the study is to explore how poetry can aid students in digesting geoscience education, and so, the chosen texts are appropriate for this task. As a recommendation however, it would be an interesting exercise for the students to take a more strictly scientific text and explore how they translate it to a poem.

*We agree with your recommendation, and we postulate this for future studies in the 'Conclusion' section.*

It would be interesting to understand how the Task Meaning themes were assigned: 'Challenging' constructed from 'Frustrating' and 'Restrictive'. Is there a guideline as to the theme nomenclature? The word "Challenging" has a connotation of a positive hurdle, whereas frustrating and restrictive connote negative feelings. This is important in identifying the subjective experience of students who may be writing poetry for the first time.

*Participants who described the task as "challenging" stated that they felt "restricted" (which is not meant to have any negative connotations), but were not frustrated, or "limited", and were frustrated. This pattern was seen throughout a lot of the replies, so "frustrating" and "restricted" were seen as offshoots of "challenging". "Restricted" was used instead of "Limited", because the latter sounds more negative and the former sounds more neutral. However, we recognise that some of the words may be ambiguous and other researchers may have created a different thematic map and used different terms as themes.*

The manuscript introduces and concludes the aim of the study as a way to explore "whether writing poetry can be used by geoscience students as an aid to their science education". For this purpose, the authors used a survey comprised of 4 questions related to the experience (Q1: How did you find the experience of writing the poem) and information-distilling process (Q2: Did writing the poem affect your engagement with the science in any way. Q3: Was a haiku an appropriate form to use?, Q4: Have you experience of writing poetry before this exercise). However, there could have been questions assessing how efficient the format was in comparison to other studying methods, since Question 3 refers to using Haiku compared to other styles of poetry. Question 4 could be followed by asking participants if they would consider employing this format in their studies at a later time.

*In the conclusion, it has been suggested that other questions, particularly open-ended ones, are explored in future studies.*

This is particularly important for supporting the aim of the study, poetry as 'an aid in education', as Participant 10 did not have prior experience with poetry and they did not find it useful for science engagement. This was not highlighted in the text.

*We thank you for your comment. We now mention in the 'Conclusion' section that future studies should take the previous experience of participants writing poetry into consideration and that Participant 10 did not have previous experience.*

I suggest reaching a pool of participants with a wider range of background experience with poetry to better explore how effective poetry is aiding students with their (geo)science education.

*We have added this suggestion for future studies in the 'Conclusion' section.*

This study serves as an introduction to how poetry could serve as an educational aid in geoscience education, but the survey to participants could be more comprehensive and better address the research question so the discussion section could go deeper to explore more insightful outcomes that have not previously been suggested elsewhere.

*We agree that this study is an introduction to the exploration of the subjective experience of science students writing poetry as an aid to their education. The limiting nature of the questions is mentioned in the 'Conclusion' section, and we now propose that future studies ask different questions that explore subjective (and objective) experience in more depth.*

**Reply to Brigid Christison**

**Thank you for taking the time to read this manuscript, and for providing helpful and specific feedback for how to improve this work. Below we have responded to all your comments and indicated how we have changed the manuscript to account for these changes (which for ease of use we have written in italics). Any line references refer to those provided in the Geoscience Communication Discussions preprint.**

This study would benefit greatly from a larger sample size, as well as a more in-depth quantitative analysis of the general trends that came out of the study. As well, I would like to see a comment on whether or not the haikus produced were accurate or informative. Though the poets themselves might say they understood and were able to capture the importance of the research they were describing, I would like the authors to comment on this themselves. This research is an important step towards recognizing that the barrier between science and art is not as firm as some might think, and that scientists could benefit from using creative thinking to better understand their research.

*The limitation of the low sample size has now been mentioned in the 'Conclusion' section, and an objective analysis is also proposed for future studies.*

*Analysing the accuracy of the haiku written by the participant was not part of the study. This study focused on the students' experience of the task and whether, in their subjective experience, they thought that it helped them. However, a sentence has been added in the conclusion: "The informative nature and accuracy of the scientific information within the haiku, as well as its effect on future academic success, may also be analysed in an objective manner.*

line 10: Specify that this study uses haikus as opposed to poetry in general

*The information in line 10 reiterates the title of this study. However, the reference to poetry later on in the abstract has been changed to "haiku".*

Lines 12-13: This implies that students benefit the most from memorising material, though I would argue that understanding or even being able to explain material is more valuable. I would think about the results of this study in terms of how the students are able to learn and understand, rather than memorise, since in your introduction you talk about the value of using poetry as a communication tool.

*We agree that the wording of the final sentence of our abstract makes it seem as though we were testing their memory. As such we have now removed "…and therefore easier to memorise efficiently" from the final sentence.*

You talk about how poetry can be used as a communication tool between disciplines, however this study attempts to gauge the efficacy of poetry writing as an education tool. I would include a paragraph discussing how the process of poetry writing can be a way for the poets to better understand the material themselves. The study by Pollack and Korol (2013) supports this argument, as you state that the students in their study were required to demonstrate "contemplation and lateral thinking," however you need to explicitly make the argument that poetry can help poets learn better, since that is the goal of your own study.

*The goal of our study is to explore the subjective experience of science students/scientists writing poetry, rather than conduct an objective analysis of whether it helps them to learn better.*

*Unfortunately, discussing the process of poetry writing would push the number of words beyond the 1500 word maximum. The results of this study, however, discuss processes involved in the task of poetry writing.*

Talk about why the process of writing poetry stimulates learning and communication. Poetry requires creative thinking, and understanding material enough to communicate it in a different type of language. It also involves sitting with material and thinking about it for a long time, something that can also help poets understand content in different ways.

*Due to the limited word count, this information is not given in the 'Introduction' section of this article. The results of this study discuss what the process of writing poetry consists of, but the reasons why this is effective is outside the focus of this study.*

Additionally, there is no comment on why the geosciences specifically were chosen. There is a long history of natural historians, including geoscientists, writing poetry. Contemplation of the natural world, and the concept of deep-time itself, can lead to intense emotional responses not easily explained via scientific language.

*The final sentence of the introduction has been extended to read "Inspired by this study, our research presents an introduction to exploring geoscience students' experience of writing academic poetry as an aid to their science education, as there is a history of both poets and scientists writing poetry on geography-related topics". The reference (Higgins, 2019) is also given to direct readers to content that discusses a history of scientists writing poetry based on geography-related topics.*

As well, plenty of literature has been written on the link between natural history and romanticism, including on the topic of geoscience poetry. I understand an in-depth historical analysis is out of the scope of this study, but it would be beneficial to touch on the historical context.

*As a GC Insights paper is limited to 500-1500 words, it is difficult to discuss contextual and historical information in as much detail as would be ideal. However, two references have been added (Higgins, 2019; Roche et al., 2018), which can be read to get some more contextual information.*

I fully agree with the comments made by Dr. Mazon. I'll add that another way of getting more participants could be reaching out to geoscience departments, conferences, and social media accounts to ask for their assistance.

*The research advertisement for this study was sent to geoscience-related university module leaders in the UK; however, to our awareness, no participants were recruited this way. Despite this, information about this form of recruitment has now been added.*

Since this is a science journal, you should separate the results and discussion sections (instead of having a single "Results & Discussion" section. The results section should present the results plainly, free from personal interpretation, and the discussion section should interpret and discuss the results in-depth".

*As a GC Insights manuscript is limited to about 500 to 1500 words, it is difficult to go into as much detail as would be ideal. Also, due to this being a qualitative study, the analysis does involve some personal interpretation. Having a results and discussion allows for a more coherent presentation of the data rather than fragmenting the information over two different sections.*

I find the structure of the results section a little disorienting. Instead of including comments in-text, I recommend bringing attention to the fact that you provide a link to the poems, and referring to them as needed. In text, it may be more effective to summarize the sentiments of each comment in-depth.

*Thank you for your suggestion. However, following on from other studies that have used poetry as a data set in this journal (e.g. Illingworth, 2020; Soldati & Illingworth, 2020) we wanted to adopt a similar formatting as we believe that it helps to present the emergent narratives of the data.*

I'd like to see your findings in graph form, grouping together common sentiments for both Task Process and Task Meaning. This would give us a quantifiable sense of the general sentiments of the participants, such as "frustrating" and "enjoyable," as they relate to the study. Graphs are also a convenient way of presenting trends, for example, I would be interested to see how the answer to question 4 (Have you experience of writing poetry before this exercise?) relates to the responses to questions 1-3.

*A GC Insights article permits only one figure as to keep information concise; therefore, it is not possible to add more figures than the one we already have. A lengthier article that is not restricted to a word limit may want to explore the trends and relationships between questions.*

Something important that is missing from this study is a discussion on the poems themselves. Did they accurately represent the science they are trying to discuss? How much information were they able to fit into the haiku format, and what information was lost? Are there parallels between the information presented in each poem, for example, do people avoid longer words or complicated concepts? You mentioned that some of the poets found the format limiting, did this come across when reading the poems? Did students prefer any of the provided text passages over the others? You include a link to the haikus themselves, so when commenting on them be sure to reference them in-text).

*An objective analysis of the poems themselves is outside the scope of this study and would extend the word limit beyond the maximum 1500 words for a GC Insights article. The Pollack and Korol (2013) study mentioned throughout goes into more detail about the accuracy of haiku written by science students.*

Line 113 – yes, but discuss how these traits (enjoyable, challenging, and valuable) relate to your research goal (determining the efficacy of poetry as a learning tool).

*The research goal of this study is to explore the subjective experience of geoscience students writing science poetry rather than to objectively determining the efficacy of this as a learning tool. The Pollock and Korol (2013) study mentioned throughout explores the efficacy and accuracy of the science poems in more rigorous detail. Future studies will aim to go into more detail. However, what we present here is novel and in line with GC Insights papers, i.e. "GC Insights present innovative and well-founded ideas related to geoscience communication in a concise way using 500–1500 words and a maximum of one figure or table. A GC Insight must be well-founded and robust, but it does not have to be explored in detail."*